# Feature Selection of Non-Dermoscopic Skin Lesion Images for Nevus and Melanoma Classification

**Felicia Anisoara Damian** [1,2], **Simona Moldovanu** [2,3], **Nilanjan Dey** [4], **Amira S. Ashour** [5] **and Luminita Moraru** [1,2,*]

1   Department of Chemistry, Physics & Environment, Faculty of Sciences and Environment, Dunarea de Jos University of Galati, 47 Domneasca Str., 800008 Galati, Romania; felicia.michis@ugal.ro
2   The Modelling & Simulation Laboratory, Dunarea de Jos University of Galati, 111 Domneasca Str., 800102 Galati, Romania; simona.moldovanu@ugal.ro
3   Department of Computer Science and Information Technology, Faculty of Automation, Computers, Electrical Engineering and Electronics, Dunarea de Jos University of Galati, 47 Domneasca Str., 800008 Galati, Romania
4   Department of Information Technology, Techno India College of Technology, West Bengal 700156, India; neelanjan.dey@gmail.com
5   Department of Electronics and Electrical Communications Engineering, Faculty of Engineering, Tanta University, Tanta 31527, Egypt; amirasashour@yahoo.com
*   Correspondence: luminita.moraru@ugal.ro; Tel.: +40-745-649-014

**Abstract:** (1) Background: In this research, we aimed to identify and validate a set of relevant features to distinguish between benign nevi and melanoma lesions. (2) Methods: Two datasets with 70 melanomas and 100 nevi were investigated. The first one contained raw images. The second dataset contained images preprocessed for noise removal and uneven illumination reduction. Further, the images belonging to both datasets were segmented, followed by extracting features considered in terms of form/shape and color such as asymmetry, eccentricity, circularity, asymmetry of color distribution, quadrant asymmetry, fast Fourier transform (FFT) normalization amplitude, and 6th and 7th Hu's moments. The FFT normalization amplitude is an atypical feature that is computed as a Fourier transform descriptor and focuses on geometric signatures of skin lesions using the frequency domain information. The receiver operating characteristic (ROC) curve and area under the curve (AUC) were employed to ascertain the relevance of the selected features and their capability to differentiate between nevi and melanoma. (3) Results: The ROC curves and AUC were employed for all experiments and selected features. A comparison in terms of the accuracy and AUC was performed, and an evaluation of the performance of the analyzed features was carried out. (4) Conclusions: The asymmetry index and eccentricity, together with $F_6$ Hu's invariant moment, were fairly competent in providing a good separation between malignant melanoma and benign lesions. Also, the FFT normalization amplitude feature should be exploited due to showing potential in classification.

**Keywords:** skin lesion; morphological operators; feature extraction; ROC curves; AUC

---

## 1. Introduction

Skin diseases vary in severity and symptoms; for example, a nevus is a benign skin lesion due to the proliferation of cells that produce pigments (melanocytes), while melanoma is a form of skin cancer with aggressive development that can develop from a benign lesion. Melanoma originates from the same pigment-producing nevus cells [1]. Both clinical and statistical types of research have confirmed that melanoma is rapidly evolving, causing high mortality among the population worldwide. In 2017, 87,110 individuals were diagnosed with melanoma in the USA [2]. The skin lesion is curable with early

stage detection, leading to the development of computer-aided detection systems for the computational diagnosis of typical skin lesions. The most significant step in such systems is feature extraction and selection, where various features, such as the asymmetry, border irregularity, and color, can be used to inspect the skin lesion pigmentation, shape, or evolution [3] or to separate melanomas from benign lesions [4]. For automated melanoma detection, numerous computer-based techniques using image segmentation [4] and support vector machines (SVMs) [5] exist. Furthermore, for quantitative and qualitative information, other features, such as color, texture, or geometrical features, can be extracted. In [3], the ABCD (A = asymmetry, B = border irregularity, C = color variegation, and D = diameter of the lesion) set and the seven-point checklist were expanded with feature E (evolving or evolutionary change) to assess the dynamic nature of skin malignancy. The reported sensitivity and specificity of the ABCDE features vary when they are used individually or in combinations. This method classified only pigmented lesions and has limited generalization capability when non-melanoma skin cancer is investigated.

Different studies have been conducted for lesion detection based on feature extraction. Korotkov and Garcia [6] reported an automatic technique for skin cancer detection on images acquired by in vivo imaging based on specific features related to pigmented skin lesions. Besides this, asymmetry, irregular edges, heterogeneity, and dynamic color features have been widely used to analyze suspicious skin lesions [7]. Smaoui and Bessassi [8] analyzed lesions based on the ABCD rule and computed the total dermatoscopic value for the malignancy detection of pigmented skin lesions. The geometric/color asymmetries, border irregularity, colors, and diameters were investigated as features. They used three skin lesions, namely, melanoma, suspicious, and benign skin lesions, in 40 samples of dermatoscopic images. The accuracy of the proposed method was 92%, with 4 false diagnoses among the 40 samples. The asymmetry of a lesion image was determined by identifying the centroids and centers of mass in each of the four lesion quadrants and using fuzzy logic techniques [9,10]. Based on the four features of asymmetry computed for each quadrant, the benign lesion and malignant melanoma discrimination results reached an accuracy of 81%. The shape features of skin lesions can be characterized by using Hu's invariant moments [11–13]. Hu's moments method uses the central moments, which integrate translation and scale normalization. The main drawback of Hu's moments consists of their redundant information. On the other hand, Hu's moments show strong differentiation abilities and are robust to noise. A different approach for shape characterization is based on spectrum analysis using fast Fourier transform (FFT) and specific Fourier descriptors. Yuan et al. [14] adopted the FFT and wavelet analysis to investigate various simple shapes and the real microstructure morphology in terms of shape similarity, axis orientation, axis ratio, and profile roughness. Similar shapes can be appropriately distinguished through FFT and Fourier descriptors.

The ABCD features are usually integrated with a large number of other features to improve the classification process, especially in multiclassification problems. In the current work, a set of necessary simple descriptors is introduced to characterize the skin lesions sufficiently to be unambiguously classified [15–18]. The asymmetry, eccentricity, circularity, asymmetry of color distribution, quadrant asymmetry, 6th and 7th Hu's moments, and FFT normalization amplitude were investigated due to their ability to distinguish the skin lesions. Receiver operating characteristic (ROC)-based feature selection approaches were employed for classification in [19]. A preprocessing step is employed for noise removal and uneven illumination reduction. The classical morphologic top-hat transform is carried out to correct the uneven illumination of the images [20,21].

Based on the above analysis, we propose this feature selection approach to use the fewest simple descriptors necessary to characterize an object adequately. Also, we aim to determine those features that can serve as good candidates for skin lesion classification. The long-term objective is to develop a teledermatology tool for the diagnosis of skin lesions and later to migrate to a mobile application.

The remaining sections introduce the segmentation process used, the mathematical approaches of the extracted features, and the proposed method in Section 2, followed by the experimental results and discussion in Section 3. Finally, Section 4 highlights the conclusions of the study.

## 2. Materials and Methods

A set of features is proposed, and these features were extracted from color digital images. Two databases with 70 melanomas and 100 nevi were investigated. The first one, denoted D1, contains raw images. The second database, D2, contains images preprocessed for noise removal and uneven illumination reduction. The feature extraction was based on the segmentation of the lesion area from the surrounding normal skin. Afterward, the area under the curve (AUC) and ROC curve were employed to ascertain the relevance of the selected features and their capability to differentiate between nevi and melanoma.

### 2.1. Preprocessing Operations

The removal of noise and artifacts due to the existence of hair is considered an important step in achieving high performance in image diagnosis. Also, uneven illumination correction is in high demand. A median de-noising filter was considered for image improvement with minimal degradation of the initial image [22]. To correct the nonuniform illumination, an algorithm using top-hat transformation was implemented [20,21]. Several examples of images processed for noise removal and nonuniform illumination correction are presented in Figure 1.

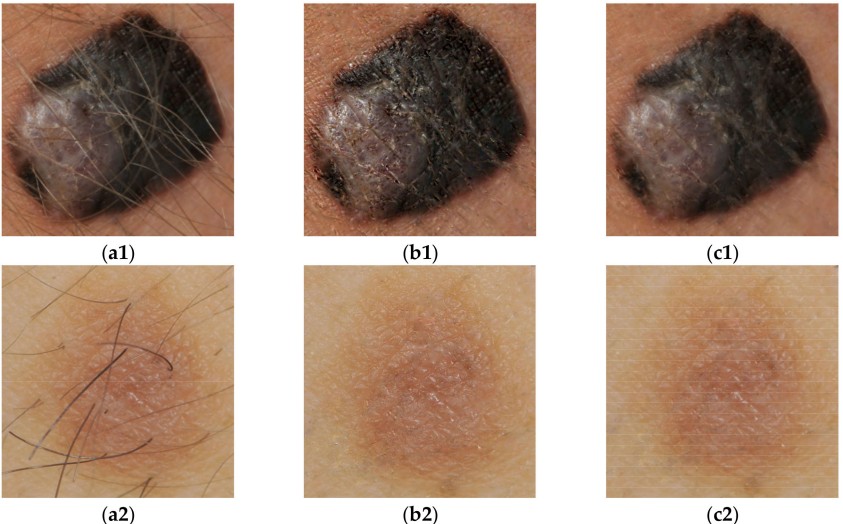

**Figure 1.** Preprocessing of a melanoma image (first raw) and nevus image (second raw): (**a1,a2**) Original image; (**b1,b2**) illumination equalization results obtained by employing classical top-hat transform; (**c1,c2**) the image given after applying the median filter.

### 2.2. Segmentation Process

Otsu's method optimally converts a grayscale image into a binary image by setting threshold T values to minimize the overlapping of the class distribution [16–18]. It is formulated as discriminatory analysis between two groups of pixel intensity level, $C_0$ and $C_1$, according to T, such that $C_0 = \{0, 1, 2, \ldots, T\}$, and $C_1 = \{T+1, T+2, \ldots, L-1\}$, where L is the image maximum gray level. Figure 2 displays the results of the segmentation process of skin lesion images.

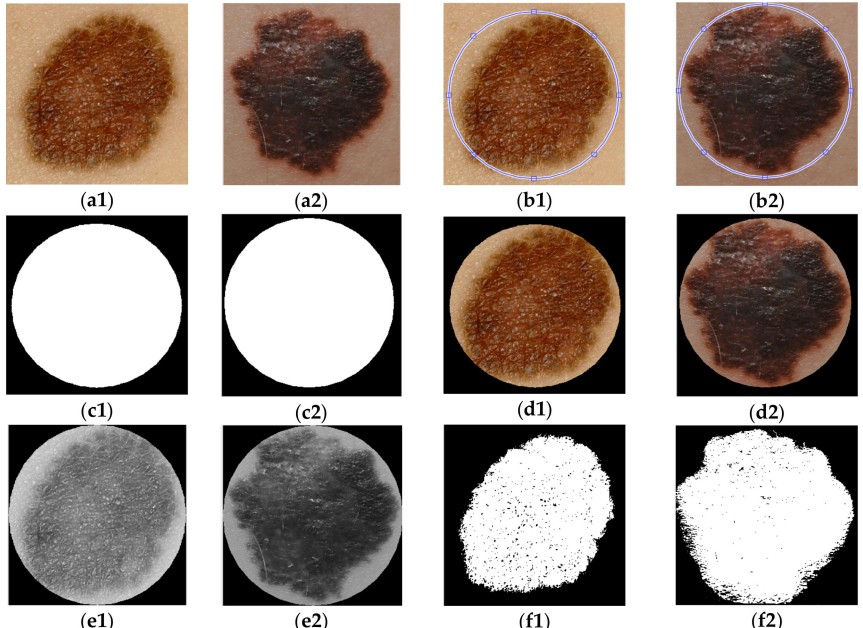

**Figure 2.** Skin lesion segmentation, with index 1 for a nevus and 2 for melanoma; (**a1,a2**) original RGB image; (**b1,b2**) ellipse circumscribing the lesion; (**c1,c2**) elliptical mask; (**d1,d2**) elliptical mask enclosing the skin lesion; (**e1,e2**) elliptical mask in grayscale containing the lesion; and (**f1,f2**) mask associated with the skin lesion (segmented image).

*2.3. Description of Features*

In the present work, various features were extracted; these have the following mathematical expressions. The asymmetry index (AI) was calculated as follows [23]:

$$AI = \frac{\Delta A_e}{A_e} \tag{1}$$

where $A_e$ is the area of the ellipse circumscribing the object, and $\Delta A_e$ is the difference between the area of the ellipse circumscribing the object and the area of the object. Besides this, the eccentricity (E) of an ellipse is the ratio of the distance from the center to the foci to the distance from the center to the vertices. It expresses the elongation degree of an ellipse/object as follows [24]:

$$E = \sqrt{1 - \frac{R_{min}}{R_{max}}} \tag{2}$$

where $R_{min}$ and $R_{max}$ are the minimum and maximum distances from the foci of the object to the contour of the object, which is delineated by a circle. A value of zero indicates a linear object. The shape analysis of skin lesions was conducted as follows: (i) the central position of the lesion (barycenter) was located using the pixel coordinates in the horizontal and vertical directions and the first-order moments $m_{0,1}$ and $m_{1,0}$; (ii) the orientation of the lesion shape was determined using the second-order moments $m_{2,0}$, $m_{1,1}$, and $m_{0,2}$ to compute the major axis orientation and the major and minor axis lengths. Only the number of pixels of the analyzed object and the elliptic contour were considered and adapted to the size regardless of the size of the skin lesion.

Accordingly, several measurements were calculated as follows.

The circularity (CIRC) excludes the local irregularities and indicates the deviation from a circular shape [25]:

$$CIRC = \frac{4\pi A}{P^2}, \tag{3}$$

where A and P are the area and perimeter of the object, respectively. Perimeter was measured by counting the number of pixel edges forming a border between pixels inside the object from the pixels outside the object, and the area as the sum of pixels inside of the object.

To measure the feature of asymmetry of color distribution (Q), the lesion was divided according to the major axis. The chi-square distance ($Q_x$) was computed using the histograms of both components, as follows [8]:

$$Q_x(e_1, e_2) = \sum_{i=1}^{N} \frac{(e_1(i) - e_2(i))^2}{e_1(i) + e_2(i)}, \ x = R, \ G, \ B, \tag{4}$$

where $e_1$ and $e_2$ are histograms of size N for each axially separated component of the lesion. This technique was applied for the R, G, and B components. The asymmetry of color distribution (Q) was computed as the average of the asymmetry scores of each component R, G, and B.

A quadrant asymmetry ($\lambda$) feature for skin lesion discrimination was used to compute a set of four new features of asymmetry. This feature is an adapted version of the method proposed in [9]. The skin lesion border mask was divided into four quadrants (each $128 \times 128$ pixels). The centroids and centers of mass of the whole lesion image and each quadrant were determined. Then, the asymmetry feature for quadrant *i* was computed as [10]

$$\lambda_i = \left( \frac{Q_k}{L_i} \right) \left( \frac{oQ_i}{oL_i} \right), \ i = \overline{1,4}, \tag{5}$$

where $k = 1$ for nevi and $k = 2$ for melanoma. $Q_i$ is the area of the object (nevus or melanoma) and $L_i$ is the area of the lesion in the particular quadrant *i*. $oQ_i$ is the distance from the centroid of the lesion within quadrant *i* to the centroid of the lesion, and $oL_i$ is the distance from the center of mass of the lesion within quadrant *i* to the center of mass of the lesion. The quadrant asymmetry ($\lambda$) was computed as the average of the quadrant asymmetry features $\lambda_i$.

Hu's moments were computed using the shape boundary information and relationships among the pixels inside the region [11–13]. Hu's seven moments contain much redundant information. Hu's invariant moments use the central moments, which integrate translation and scale normalization. The moments $F_1$ to $F_6$ are translation, size, and rotation invariant. $F_7$ is skew invariant. We analyzed in our approach only the moments $F_6$ and $F_7$:

$$F_6 = (\eta_{20} - \eta_{02})((\eta_{30} + \eta_{12})^2 - (\eta_{21} + \eta_{03})^2) + 4\eta_{11}(\eta_{30} + \eta_{12})(\eta_{21} + \eta_{03}), \tag{6}$$

$$F_7 = (3\eta_{21} - \eta_{03})(\eta_{21} + \eta_{03})(3(\eta_{30} + \eta_{12})^2 - (\eta_{21} + \eta_{03})^2) - (\eta_{30} - 3\eta_{12})(\eta_{21} + \eta_{03})(3(\eta_{30} + \eta_{12})^2 - (\eta_{21} + \eta_{03})^2), \tag{7}$$

where $\eta_{ij}$, $i + j \geq 2$ denotes invariance concerning both translation and scale. The major weakness of Hu's theory is that it does not provide for the possibility of any generalization.

The FFT normalization amplitude feature computed as a Fourier transform descriptor is introduced to analyze the shape characteristics of lesions. The discrete form of the fast Fourier transform (FFT) is [6]

$$F(p, q) = \sum_{m=0}^{M-1} \sum_{n=0}^{N-1} f(m, n) e^{-j\frac{2\pi mp}{M}} e^{-j\frac{2\pi nq}{N}} \tag{8}$$

where $p = 0, 1, M - 1$; $q = 0, 1, N - 1$. According to FFT analysis, the Fourier amplitude can be written as follows:

$$F(u) = R(u) + jI(u), \tag{9}$$

where R(u) and I(u) are the real and imaginary parts of the transforming data; j is the square root of $-1$; $u = 1, 2, \ldots, N - 1$. The FFT normalization amplitude acts as an analyzed feature and contains shape orientation. It is as follows:

$$FFT = \frac{\|F_u\|}{\|F_0\|} \tag{10}$$

where ‖F$_u$‖ indicates the norm of the amplitude of F(u) or the Fourier spectrum, and ‖F$_{u0}$‖ is the amplitude of the 0th Fourier descriptor.

Finally, the AUC and the ROC curve were used to estimate the effectiveness of a selected feature by comparison with a perfect classifier (a perfect classifier has an AUC of 100%). If AUC < 0.6, the selected features are irrelevant for the proposed goal. ROC plots were made of the sensitivity versus the false alarm rate (i.e., 1—specificity). The sensitivity or the detection probability was computed as TP/(TP + FN) and the specificity or false alarm probability was TN/(TN + FP), where TP is true positive, FN is false negative, TN is true negative, and FP is false positive.

### 2.4. The Proposed Feature Selection Method

These steps of the proposed method are illustrated in Figure 3.

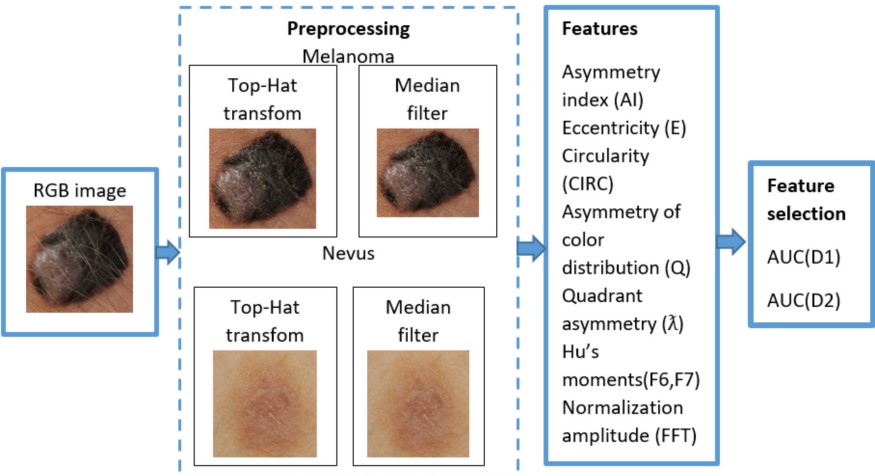

**Figure 3.** Proposed feature selection method.

## 3. Results and Discussion

To evaluate the performance of the proposed feature selection procedure, 170 color digital images (70 melanomas and 100 nevi) from the Digital Archive of the Department of Dermatology of the University Medical Center in Groningen (UMCG) were used [26]. In this study, the programming environment was MATLAB R2018a. The hardware experimental environment consisted of an Intel (R) Core (TM) i3-4030U x64-based processor, 1.9 GHz CPU, 4 GB installed memory (RAM), and 64-bit operating system.

The present work is based on the features selected using the proposed procedure to differentiate between benign nevi and melanoma lesions. The proposed features describe the structure of the skin lesions and the appearance differences between benign and malignant. A total of eight features were quantified for each lesion type and dataset. The ROC curves and AUC are illustrated in Figure 4. Table 1 shows the average values of the performance measures

The area under the curve (AUC) for AI (asymmetry index) is 0.81 for raw images and 0.89 for the processed images, showing very good separation between the malignant melanoma and benign lesions. Similarly, for feature E (eccentricity), the AUC values are 0.75 (D1) and 0.77 (D2), indicating good separation between malignant melanoma and nevi. The CIRC, Q, and λfeatures show values of area under the ROC plot ranging from 0.51 to 0.58, indicating modest separation of the two classes. The FFT normalization amplitude has an AUC of 0.64 for raw images and 0.69 for images in dataset D2 and presents average separation of the two classes. F$_6$ and F$_7$ Hu's invariant moments have AUC values of 0.71(D1)/0.77(D2) and 0.61(D1)/0.66(D2), respectively. F$_6$ has good separation, while F$_7$ has average separation between malignant melanoma and benign lesions.

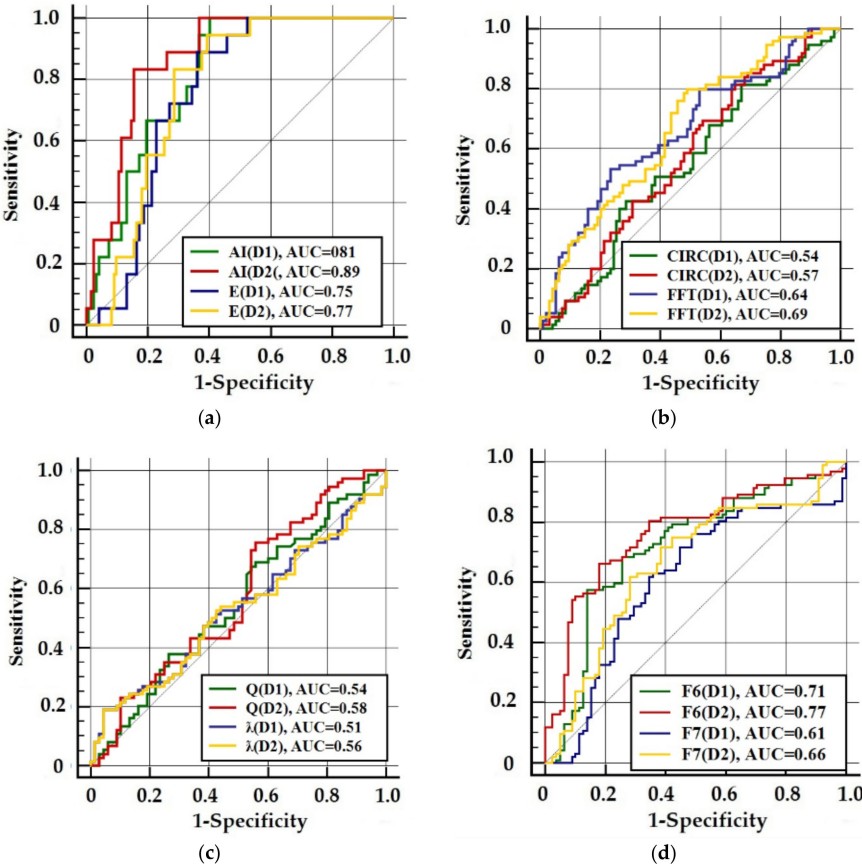

**Figure 4.** Receiver operating characteristic (ROC) curves and area under the curve (AUC) for discriminating melanoma from nevi in the D1 and D2 datasets: (**a**) AI (asymmetry index) and E (eccentricity) features; (**b**) circularity (CIRC) and FFT normalization amplitude features; (**c**) asymmetry of color distribution (Q) and quadrant asymmetry ($\lambda$) features; (**d**) $F_6$ and $F_7$ Hu's invariant moments.

**Table 1.** The sensitivity, specificity, and accuracy of the analyzed features for the original (D1) and processed (D2) images.

|  | AI Feature | E Feature | CIRC Feature | FFF Feature | Q Feature | $\lambda$ Feature | $F_6$ Feature | $F_7$ Feature |
|---|---|---|---|---|---|---|---|---|
| Sensitivity (D1) | 0.98 | 0.90 | 0.81 | 0.53 | 0.61 | 0.67 | 0.68 | 0.76 |
| Sensitivity (D2) | 0.84 | 0.94 | 0.81 | 0.80 | 0.63 | 0.65 | 0.72 | 0.62 |
| Specificity (D1) | 0.64 | 0.66 | 0.32 | 0.76 | 0.57 | 0.46 | 0.80 | 0.51 |
| Specificity (D2) | 0.87 | 0.63 | 0.35 | 0.51 | 0.58 | 0.53 | 0.82 | 0.71 |
| Accuracy (D1) | 0.82 | 0.78 | 0.57 | 0.65 | 0.59 | 0.57 | 0.74 | 0.64 |
| Accuracy (D2) | 0.85 | 0.79 | 0.58 | 0.66 | 0.61 | 0.59 | 0.77 | 0.67 |

In terms of sensitivity and specificity, the asymmetry index, for images in D2, shows balanced accuracy of correct classification for melanoma and nevi (Table 1). Conversely, eccentricity and the FFT normalization amplitude (for dataset D2) indicate sensitivity of 0.94/0.80 versus specificity of 0.63/0.51 and accuracy of 0.79/0.66, respectively. The eccentricity and FFT normalization amplitude are relevant features for melanoma diagnosis. $F_6$ Hu's invariant moment is important for nevus diagnosis as its specificity/sensitivity is 0.82/0.72 with accuracy of 0.772. $F_7$ Hu's invariant moment follows the same trend, but the accuracy is 0.67 (D2). Asymmetry of color distribution (Q) and quadrant asymmetry ($\lambda$) features showed no significant contribution as assessed by the sensitivity, specificity, and accuracy.

According to these results, two dominant uncorrelated features, AI (asymmetry index) and E (eccentricity), are good candidates for skin lesion classification. They are geometric features and indicate that melanoma usually shows important disturbances in terms of symmetry and regular geometric shapes, and they can thus be discriminated from benign lesions. Also, the FFT normalization amplitude can act as a descriptor of the skin lesion malignancy as it is built using the shape characteristics

of lesions. On the other hand, the quadrant asymmetry (λ) feature failed to be a relevant feature for classification. When the analysis is performed partially in quadrants, some information is lost and the lesion discrimination results are not satisfactory. The asymmetry of color distribution (Q) describes the biaxial asymmetry in terms of color distribution and shows inferior performance in comparison with geometric features, despite the increased complexity of the extraction algorithm. However, this feature has potential in the evaluation of malignancy when a set of features is generated by combining the color and geometric features. The Hu's moments characterize global features and are appropriate for shape recognition. The $F_6$ moment is translation, size, and rotation invariant and $F_7$ is skew invariant. $F_6$ is more highly recommended for classification tasks than $F_7$ but the computation process of both features is quite complicated. For all proposed features, there were better results for processed images (D2).

Moreover, a comparison with previous studies that used the same dataset was conducted, where Giotis et al. [26] proposed a classification system based on color and texture lesion descriptors using the same image database. The automatically extracted features were combined with a set of visual attributes provided by the examining physician. The most valuable features were selected using a majority vote, and 81% diagnostic accuracy was reported. The cost of the computation process was high, where the features were generated from images and visual features. Almaraz-Damian et al. [27] proposed a computer-aided detection system for the detection and classification of melanoma skin lesions. They used a fusion technique to concatenate the features provided by the ABCD rule and a deep learning algorithm and used a mutual information metric to extract relevant information among features. Then, some classifiers such as logistic regression, support vector machine with linear and radial basis function (RBF) kernel, and vector machine algorithms were employed. The proposed method achieved an accuracy of 92.40% but is computationally expensive. Accordingly, due to the effectiveness of the proposed approach, it is suggested to compare the proposed approach with other feature selection methods for benchmarking and other previous studies on dermoscopic images, such as in [28–32]. In the future, we intend to migrate this method to a mobile application.

## 4. Conclusions

In this study, a set of eight features were investigated in terms of their potential to differentiate melanoma from nevi. These features are devoted to asymmetry evaluation based on shape and color assessment. Emphasis was placed on the evaluation of lesion shape and color, which are expressed by various mathematical approaches. In an attempt to break out of the patterns of the ABCD rule, we investigated an atypical feature, namely, the FFT normalization amplitude computed as a Fourier transform descriptor that focuses on geometric signatures of the skin lesion using frequency domain information. The asymmetry index and eccentricity, together with $F_6$ Hu's invariant moment, are fairly competent in providing good separation between malignant melanoma and benign lesions. Also, the FFT normalization amplitude feature should be exploited as it shows potential in classification. Additionally, the AI (asymmetry index) feature is competitive for multiclass classification (melanoma and nevus skin lesions), eccentricity and FFT normalization amplitude for melanoma diagnosis, and $F_6$ and $F_7$ Hu's invariant moments for nevus diagnosis.

**Author Contributions:** L.M.: Research project—Conception, Organization, and Execution; Statistical Analysis—Design, Execution, Review, and Critique; Manuscript—Writing of the first draft, Review, and Critique. F.A.D.: Research project—Organization and Execution; Statistical Analysis—Design, Review, and Critique. S.M.: Research project—Conception, Organization and Execution; Software validation. N.D.: Research project—Software validation; Manuscript—Review and Critique. A.S.A.: Research project—statistical/results analysis, interpretation, and Critique, Manuscript—writing, review and Critique. All authors have read and agreed to the published version of the manuscript.

**Funding:** This research received no external funding.

**Conflicts of Interest:** The authors declare no conflict of interest.

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
