# Peer review of "Feature Selection of Non-Dermoscopic Skin Lesion Images for Nevus and Melanoma Classification"

_computation, doi:10.3390/computation8020041_

Round 1

Reviewer 1 Report

The authors present a feature selection method for nevi and melanomas classification.

The overall methodology is quite simple and questionable to be applicable to such a demanding machine learning problem.

  • The segmentation method is based on Otsu's threshold, and texture features are derived from first-order statistics without any image pre-processing. The data set used has images interrupted by noise (appearance of hair, glossy due to high reflectance, uneven illumination, etc)
  • Geometrical descriptors are too limited and simplistic. Overall, they encode the same information: the deviation from a circular shape and not the contour irregularity (indentations, fractal-dimension). (Eq 13 and eq 14 measure the same thing!)
  • Feature selection methodology applies metrics to rank the features' 'ability' in differentiating the two classes, and finally, authors result in a single descriptor. However, it is hardly rare to achieve two-class classification with a single descriptor, and moreover, such feature selection techniques are not robust and are leading to the inherent problem of nesting.

Authors use the terms Sensitivity and Specificity for both melanoma and nevi (Figure 3 !); In a two-class problem such as benign (nevi) versus malignant(melanoma), sensitivity and specificity refer to the classification accuracies of melanoma and nevi respectively. Also, applying 10 fold cross-validation, results should be given in terms of mean Sensitivity and mean Specificity.

Authors use a rather small dataset (170 images) of non-dermoscopic images of skin lesions, without reasoning for their choice.
In bibliography there is continuing research on large datasets of dermoscopy images:

https://onlinelibrary.wiley.com/doi/full/10.1111/srt.12622
https://pubmed.ncbi.nlm.nih.gov/31017580
https://www.isic-archive.com/

:

Author Response

Authors’ response: We are thankful to the respected reviewer for this encouraging comment and the precious time to provide the feedback. We response to the reviewer's comments as follows.

(R1.1): The segmentation method is based on Otsu's threshold, and texture features are derived from first-order statistics without any image pre-processing. The data set used has images interrupted by noise (appearance of hair, glossy due to high reflectance, uneven illumination, etc)

Authors’ response: We are appreciating this comment. We would like to clarify that the main target of the proposed method is to develop an efficient feature selection method even in the case of noisy images. In addition, the images of the used dataset have good contrast. Accordingly, we have mentioned in Page 2 (lines 94-95) that:

“We are devoted to the case when the preprocessing step, i.e. denoising, hair removal, uneven illumination correction, etc.) is avoided, where the used dataset has good contrast.”

In addition, we have recommended including the pre-processing step in the future work revealing in Page 2 (Lines 95-98) that:

“Due to the effectiveness of the proposed method even without dermoscopy images’ denoising, it is recommended to apply different pre-processing stages in the future work, including hair removal, illumination and contrast enhancement before applying the proposed feature selection procedure.”

(R1.2):  Geometrical descriptors are too limited and simplistic. Overall, they encode the same information: the deviation from a circular shape and not the contour irregularity indentations, fractal-dimension). (Eq 13 and eq 14 measure the same thing!)

Authors’ response: We are thankful to this comment. Generally, our goal is to use the fewest necessary uncomplex descriptors to characterize a skin lesion adequately so that it may be unambiguously classified. We used the compactness because the objects having irregular and complicated boundaries, like melanoma, will have decreased compactness value. It is used as an indicator for the boundary irregularity. Circularity excludes local irregularities and indicates the deviation from a circular shape. By using these two features, we are addressing the boundary irregularity criterion with the goal of increasing the lesion discrimination compared to the use of each feature alone. Accordingly, we mentioned in Page 4 (lines 144-152) that:

“The shape analysis of skin lesion was conducted as follows: (i) the central position of lesion (barycenter) is located using the pixel co-ordinates in the horizontal and vertical directions and the first-order moments m0,1 and m1,0; (ii) the orientation of lesion shape is determined using the second-order moments m2,0, m1,1 and m0,2 to compute the major axis orientation and the major and minor axis lengths. Only the number of pixels of the analyzed object and the elliptic contour are considered and adapted to the size regardless the size of the skin lesions. Accordingly, several measurements are calculated as follows. The circularity excludes the local irregularities and indicates the deviation from a circular shape.”

Also, we clarified in Page 5 (Lines 157-159) that:

By using CIRC and COMP features we are addressing the boundary irregularity criterion with the goal of increasing the lesion discrimination compared to the use of each feature alone.”

(R1.3): Feature selection methodology applies metrics to rank the features' 'ability' in differentiating the two classes, and finally, authors result in a single descriptor. However, it is hardly rare to achieve two-class classification with a single descriptor, and moreover, such feature selection techniques are not robust and are leading to the inherent problem of nesting.

Authors’ response: We are appreciating the respected reviewer for this valuable comment. We have performed a new experiment using machine learning to ensure the best number of the selected features as explained in Page 6 (Line 182-197) revealing that:

  “To ensure the best number of the selected features, an artificial neural network (ANN) model is proposed to recognize the capability of features in V1 (including 6 features), V2 (including 3 features), and V3 (including 2 features) to distinguish the benign or malignant classes vs. the declared goal to reduce the data high dimensional [25].  An ANN of 10 hidden layers using the Scaled Conjugate Gradient procedure for training is applied. Then, the accuracy is used as quantitative and qualitative measures to compare the performance of the feature vectors.

Also, in Page 8 (Line 241-250)

In addition, the confusion matrices for ANN classification using the different three selected feature vectors separately, namely V1, V2, or V3 are illustrated in Figure 5.

(a)

(b)

(c)

Figure 5. The confusion matrices for ANN classification using (a) V1, (b) V2, and (c) V3, respectively.

Furthermore, Table 3 shows the classification results of the ANN using V1, V2, or V3 to classify the nevi and melanomas classes.

Table 3. ANN accuracy performance over the images in the dataset using different number of the selected features

Classification Accuracy

V1 (6 features)

V2 (3 features)

V3 (2 features)

77.5%

100%

64%

Figure 5 and Table 3 illustrate that the feature vector V2 provides the highest accuracy. This result supports the aim for dimensionality reduction without loss of the significant information by using only three features.

Also, in Page 9 (Line 261-272)

The classification performance indicates that using only one selected feature as a best descriptor disables the classifier to differentiate between benign nevi and melanoma lesions efficiently. In an attempt to find the balance between the requirement to have a reasonable data dimension and to speed-up the analysis and to maintain the accuracy and efficacy of the investigation reasonable, only the most significant selected features in V2 are considered the best solution. This guarantees the balance between the better performance and the desiderata of reduced number of features. Therefore, the best performance is achieved for  despite of the results provided by CV analysis, which indicated the AI –based features are the most relevant features to distinguish melanoma from nevi with minimum of CV =0.68. Accordingly, the proposed approach recommended the use of AI classifier after selecting the most significant features as a second step for final selection of the imperative features.”

(R1.4): Authors use the terms Sensitivity and Specificity for both melanoma and nevi (Figure 3 !); In a two-class problem such as benign (nevi) versus malignant(melanoma), sensitivity and specificity refer to the classification accuracies of melanoma and nevi respectively. Also, applying 10 fold cross-validation, results should be given in terms of mean Sensitivity and mean Specificity.

Authors’ response: We are thankful to this comment. We have modified the paper accordingly in Page 7 (Figure 3).

 (R1.5): Authors use a rather small dataset (170 images) of non-dermoscopic images of skin lesions, without reasoning for their choice. In bibliography there is continuing research on large datasets of dermoscopy images:

https://onlinelibrary.wiley.com/doi/full/10.1111/srt.12622
https://pubmed.ncbi.nlm.nih.gov/31017580
https://www.isic-archive.com/

Authors’ response:  We are appreciating this comment; however, we intended to highlight this dataset which includes non-dermoscopic images which we interested to work on, while the other datasets include dermoscopic images. In addition, we would like to clarify that the size of the used dataset meets the general requirements, where according to She et al., the sample size should meet the following criterium: Q > 3cp, where Q denotes the sample size, c is the number of classes (c = 2, nevi and melanoma) and p the number of features. We have initially proposed p = 6 features. So, Q = 36 sample size, as a minimal request. Hence, the 170 images which we used meet this requirement.

She, Z., Liu, Y., & Damatoa, A. (2007) ‘Combination of features from skin pattern and ABCD analysis for lesion classification’. Skin Research & Technology, 13(1), 25–33

Reviewer 2 Report

The autor should benchmark there technique and sould includ the results. 

Selection of the technique is not clear and i am do not see its greater advantage. 

Author Response

(R2.1): The author should benchmark there technique and should include the results. 

Authors’ response: We are appreciating this comment. We have performed a new experiment using machine learning to ensure the best number of the selected features as explained in Page 6 (Line 192-197) revealing that:

  “To ensure the best number of the selected features, an artificial neural network (ANN) model is proposed to recognize the capability of features in V1 (including 6 features), V2 (including 3 features), and V3 (including 2 features) to distinguish the benign or malignant classes vs. the declared goal to reduce the data high dimensional [25].  An ANN of 10 hidden layers using the Scaled Conjugate Gradient procedure for training is applied. Then, the accuracy is used as quantitative and qualitative measures to compare the performance of the feature vectors.

Also, in Page 8 (Line 241-250)

In addition, the confusion matrices for ANN classification using the different three selected feature vectors separately, namely V1, V2, or V3 are illustrated in Figure 5.

(a)

(b)

(c)

Figure 5. The confusion matrices for ANN classification using (a) V1, (b) V2, and (c) V3, respectively.

Furthermore, Table 3 shows the classification results of the ANN using V1, V2, or V3 to classify the nevi and melanomas classes.

Table 3. ANN accuracy performance over the images in the dataset using different number of the selected features

Classification Accuracy

V1 (6 features)

V2 (3 features)

V3 (2 features)

77.5%

100%

64%

Figure 5 and Table 3 illustrate that the feature vector V2 provides the highest accuracy. This result supports the aim for dimensionality reduction without loss of the significant information by using only three features.

Also, in Page 9 (Line 261-272)

The classification performance indicates that using only one selected feature as a best descriptor disables the classifier to differentiate between benign nevi and melanoma lesions efficiently. In an attempt to find the balance between the requirement to have a reasonable data dimension and to speed-up the analysis and to maintain the accuracy and efficacy of the investigation reasonable, only the most significant selected features in V2 are considered the best solution. This guarantees the balance between the better performance and the desiderata of reduced number of features. Therefore, the best performance is achieved for  despite of the results provided by CV analysis, which indicated the AI –based features are the most relevant features to distinguish melanoma from nevi with minimum of CV =0.68. Accordingly, the proposed approach recommended the use of AI classifier after selecting the most significant features as a second step for final selection of the imperative features.”

Furthermore, we recommended comparing our proposed approach with other feature selection methods in the future in Page 9 (lines 290-291) revealing that:

“Furthermore, it is recommended to compare the proposed approach with other feature selection methods for benchmarking.”

(R2.2): Selection of the technique is not clear and i am do not see its greater advantage. 

Authors’ response: We are thankful to the respected reviewer; however, we would like to clarify to avoid the extensive search using the traditional feature selection procedures by using our two-level selection method using few measurements. To clarify the impact of our approach, we revealed in Page 2 (lines 81-93) that:

“The ABCD features are usually integrated with a large number of other features to improve the classification process, especially in the multi-classification problems. In the current work, a set of necessary simple descriptors are introduced to characterize the skin lesions adequately to be unambiguously classified [11-14]. The eccentricity, circularity, compactness, skewness, and kurtosis are investigated as part of the feature vector to distinguish the skin lesions to enhance the classification performance. For skin lesion classification, we have used individual features and various combinations of them sequentially using the following procedure. In the first phase, the dispersion index ID showing the scatter distribution of pixels in an object is considered as a feature selector for dimensionality reduction [15]. In the second phase, the Youden (J) index and the ROC-based feature selection approaches are employed for classification [16]. Afterward, to reduce the redundancy between features (or correlated features), 10-fold cross-validation method is applied [17]. Then, an ANN model is proposed to recognize the capability of selected features to distinguish the benign or malignant classes vs. the declared goal to reduce the data high dimensional.”

In addition, we mentioned in Page 3 (lines 99-103) that:

“In view of the above analysis, we proposed this feature selection approach with the goal to use the fewest necessary simple descriptors to characterize an object adequately so that it may be unambiguously classified aiming to determining whether a small number of features can serve as a diagnostic classifier to distinguish nevi from melanoma. The long-term objective is to develop a tele-dermatology tool for the diagnosis of skin lesion and later to migrate to a mobile application.” 

Reviewer 3 Report

Authors proposed a classification scheme for classifying benign and malignant lesions. The article is well-written with some mistakes. My comments are as follows.

  1. Authors have mentioned related work but did not critically analyse it. Authors should do a critical analysis like what is missing in existing work and why they are proposing their classification scheme.
  2. From the references, it seems that the authors didn’t review the recent work done in skin lesion classification. The most updated reference authors mentioned is from 2017. Authors should read recent papers and then do their analysis.  
  3. On line 71, authors should mention what value of K is used(although they have mentioned it later on).
  4. In the segmentation section, authors have just written the Otsu’s method and its equations which are well-known. Only reference will be enough as it is a well-established method. Authors should show some experimental results in this section which shows segmentation results on different types of images(some benign and some malignant). Although, authors mentioned results of one image in Figure 2, however, they should show results on a variety of images.
  5. Line 144: authors didn’t explain CIRC. Authors should explain it first and then mentioned how it is calculated.
  6. Line 168: authors have just extracted AI, S, K, COMP, CIRC and E. Is it’s a large number of features?
  7. What is the size of the elliptic contour? Is it same for all images? As the size of the benign and malignant lesions may differ. I suggest authors to explain this in detail.
  8. For classification, authors have just mentioned that they have used 10-fold cross-validation method, however, they didn’t mention which classifier they have used.
  9. Figure 4: spelling mistake:

    EXCCENTRICITY -> eccentricity  and ASYMETRY -> asymmetry

  10. Authors have used a well-known dataset; however, they didn’t compare their results with existing studies who have used the same dataset.
  11. Figure 3: ROC curve

x-axis title is not correct. It should be 1- Specificity

why x-axis values are from 0-80 why not 0-100? 

x-axis and y-axis values should be from 0-1 not from 0-100.

In AUC and J values: . should be used instead of ,

Author Response

article is well-written with some mistakes. My comments are as follows.

(R3.1): Authors have mentioned related work but did not critically analyse it. Authors should do a critical analysis like what is missing in existing work and why they are proposing their classification scheme.

Authors’ response: We are appreciating this comment. We have included a new related work ‘[10]’ and improved our related work by including their analysis in Page 2 (lines 58-80) as follows:

“Ramezani et al. [9] extracted 187 features including the asymmetry, contour irregularity, and color variation, to distinguish between benign and malignant types of the pigmented skin lesions in macroscopic images. Then, the principle component analysis (PCA) was used for data reduction and features selection, where 13 optimal features were selected. The image dataset was collected from several online dermatology atlases. The threshold-based segmentation was applied followed by the classification process with accuracy 82.2%. However, the cost of the computational process is high as large number of relevant features (13 features) was selected.

In [3], the ABCD (A = asymmetry, B = border irregularity, C = colour variegation and D = diameter of lesion) set and the 7-point checklist were expanded with E (Evolving or evolutionary change) feature to assess the dynamic nature of skin malignancy. The reported sensitivity and specificity of the ABCDE features vary when they are used individually or in combinations. This method classified only pigmented lesions and has limited generalization capability when non-melanoma skin cancer was investigated. In [4], the ABCD criteria, LBP& GLCM and an ANN-back propagation neural network classifier were used to classify the benign or malignant stage of skin lesions. However, this approach suffered from the high computational load. Apart from the reported results in [3] and [4], our approach expands the standard ABCD features for skin lesion classification by introducing a set of simple descriptors to characterize the skin lesions adequately to enhance the classification performance.

The results reported in [10] were obtained by combining the textural and color features of the skin lesion and a Multilayer Feed-Forward ANN for classification. The ABCD features were fed to the ANN for performance analysis, while the GLCM and the statistical color analysis were used for training and testing phases. A modified standard deviation coupled with the ABCD features increases the melanoma detection with 93.7% accuracy.” 

Added reference:

[10] Muniba Ashfaq, Nasru Minallah, Zahid Ullah, Arbab Masood Ahmad, Aamir Saeed, Abdul Hafeez, Performance analysis of low-level and high-level intuitive features for melanoma detection, Electronics 2019, 8, pp. 672 (20 pages)

(R3.2): From the references, it seems that the authors didn’t review the recent work done in skin lesion classification. The most updated reference authors mentioned is from 2017. Authors should read recent papers and then do their analysis.

Authors’ response: The authors appreciate the reviewer’s comment. We have updated the references list by recent publications, where the included references as follows.

Added references:

  1. Monisha, M., Suresh, A., Bapu, B.R.T. et al. Classification of malignant melanoma and benign skin lesion by using back propagation neural network and ABCD rule, Cluster Comput, 2019, 22, pp. 12897–12907.
  2. Celebi ME, Codella N, Halpern A. Dermoscopy Image Analysis: Overview and Future Directions. IEEE J Biomed Health Inform. 2019; 23(2), pp.474-478.
  3. Ashfaq M, Minallah N, Ullah Z, Ahmad AM, Saeed A, Hafeez A, Performance analysis of low-level and high-level intuitive features for melanoma detection, Electronics 2019, 8(6), pp. 672 (20 pages).
  4. Kokonendji CC, Puig P. Fisher dispersion index for multivariate count distributions: A review and a new proposal. J Multivar Anal. 2018; 165, pp. 180-193.
  5. Diana SBR. Skin cancer detection and stage prediction using image processing techniques. Int J Eng Technol. 2018; 7(1), pp. 204-209.
  6. Xie F., Fan H., Li Y., Jiang Z, Meng R., Bovik A. Melanoma classification on dermoscopy images using a neural network ensemble model. IEEE Transactions on Medical Imaging. 2016; 36(3), pp. 849-858.
  7. Hosny, KM, Kassem MA, Foaud MM. Classification of skin lesions using transfer learning and augmentation with alex-net. PloS one; 14(5), e0217293 (2019)

(R3.3): On line 71, authors should mention what value of K is used (although they have mentioned it later on).

Authors’ response: We are appreciating this comment. We used 10-fold cross-validation to evaluate the classification performance of the classifiers (K = 10) as we revealed in Page 6 (Lines 189-190).

(R3.4): In the segmentation section, authors have just written the Otsu’s method and its equations which are well-known. Only reference will be enough as it is a well-established method. Authors should show some experimental results in this section which shows segmentation results on different types of images (some benign and some malignant). Although, authors mentioned results of one image in Figure 2, however, they should show results on a variety of images.

Authors’ response: We are thankful to this comment. We have modified Section (2.1) by deleting the well-established method. In addition, we added some experimental results of the segmentation results on different types of images (benign and malignant) are presented in Figure 1 (below).

 (a1)

(a2)

(b1)

(b2)

(c1)

(c2)

(d1)

(d2)

(e1)

(e2)

(f1)

(f2)

Figure 1. Skin lesions segmentation, index 1 for nevus and 2 for melanoma; (a1), (a2) original RGB image; (b1), (b2) ellipse circumscribed to the lesion; (c1), (c2) elliptical mask; (d1), (d2) elliptical mask enclosing the skin lesion; (e1), (e2) elliptical mask in the gray scale containing the lesion, and (f1), (f2) mask associated with skin lesion (segmented image).

(R3.5): Line 144: authors didn’t explain CIRC. Authors should explain it first and then mentioned how it is calculated.

Authors’ response: The authors appreciate the reviewer’s comment. We have updated the paragraph in the revised paper Page 4 (line 151), as circularity (CIRC).

(R3.6): Line 168: authors have just extracted AI, S, K, COMP, CIRC and E. Is it’s a large number of features?

Authors’ response: The authors appreciate the reviewer’s comment. We have updated the paragraph in the revised paper (lines 99-103) as follows: “In view of the above analysis, we proposed this feature selection approach with the goal to use the fewest necessary simple descriptors to characterize an object adequately so that it may be unambiguously classified aiming to determining whether a small number of features can serve as a diagnostic classifier to distinguish nevi from melanoma. The long-term objective is to develop a tele-dermatology tool for the diagnosis of skin lesion and later to migrate to a mobile application.” 

(R3.7): What is the size of the elliptic contour? Is it same for all images? As the size of the benign and malignant lesions may differ. I suggest authors to explain this in detail.

Authors’ response: The authors thank the reviewer for this observation. Generally, we compute the best fit ellipse to the shape for the non-dermoscopic RGB image. Then, we followed the following shape analysis procedure of the skin lesion was conducted as follows:

  1. The central position of lesion (barycenter) is located using the pixel co-ordinates in the horizontal and vertical directions and the first-order moments m0,1 and m1,0;
  2. The orientation of lesion shape is determined using the second-order moments m2,0, m1,1 and m0,2 to compute the major axis orientation and the major and minor axis lengths.

Only the number of pixels of the analyzed object and the elliptic contour are involved and adapted to the size regardless the size of the skin lesions. Accordingly, we have provided the required details in (lines 144-149) revealing that:

“The shape analysis of skin lesion was conducted as follows: (i) the central position of lesion (barycenter) is located using the pixel co-ordinates in the horizontal and vertical directions and the first-order moments m0,1 and m1,0; (ii) the orientation of lesion shape is determined using the second-order moments m2,0, m1,1 and m0,2 to compute the major axis orientation and the major and minor axis lengths. Only the number of pixels of the analyzed object and the elliptic contour are considered and adapted to the size regardless the size of the skin lesions.”

(R3.8): For classification, authors have just mentioned that they have used 10-fold cross-validation method, however, they didn’t mention which classifier they have used.

Authors’ response: We are thankful to the respected reviewer. We have performed an experiment using machine learning to ensure the best number of the selected features as explained in Page 6 (Line 192-197) revealing that:

  “To ensure the best number of the selected features, an artificial neural network (ANN) model is proposed to recognize the capability of features in V1 (including 6 features), V2 (including 3 features), and V3 (including 2 features) to distinguish the benign or malignant classes vs. the declared goal to reduce the data high dimensional [25].  An ANN of 10 hidden layers using the Scaled Conjugate Gradient procedure for training is applied. Then, the accuracy is used as quantitative and qualitative measures to compare the performance of the feature vectors.

Also, in Page 8 (Line 241-250) we revealed that:

In addition, the confusion matrices for ANN classification using the different three selected feature vectors separately, namely V1, V2, or V3 are illustrated in Figure 5.

(a)

(b)

(c)

Figure 5. The confusion matrices for ANN classification using (a) V1, (b) V2, and (c) V3, respectively.

Furthermore, Table 3 shows the classification results of the ANN using V1, V2, or V3 to classify the nevi and melanomas classes.

Table 3. ANN accuracy performance over the images in the dataset using different number of the selected features

Classification Accuracy

V1 (6 features)

V2 (3 features)

V3 (2 features)

77.5%

100%

64%

Figure 5 and Table 3 illustrate that the feature vector V2 provides the highest accuracy. This result supports the aim for dimensionality reduction without loss of the significant information by using only three features.

Also, we mentioned in Page 9 (Line 262-272) that:

“The classification performance indicates that using only one selected feature as a best descriptor disables the classifier to differentiate between benign nevi and melanoma lesions efficiently. In an attempt to find the balance between the requirement to have a reasonable data dimension and to speed-up the analysis and to maintain the accuracy and efficacy of the investigation reasonable, only the most significant selected features in V2 are considered the best solution. This guarantees the balance between the better performance and the desiderata of reduced number of features. Therefore, the best performance is achieved for  despite of the results provided by CV analysis, which indicated the AI –based features are the most relevant features to distinguish melanoma from nevi with minimum of CV =0.68. Accordingly, the proposed approach recommended the use of AI classifier after selecting the most significant features as a second step for final selection of the imperative features.”

(R3.9): Figure 4: spelling mistake: EXCCENTRICITY -> eccentricity  and ASYMETRY -> asymmetry

Authors’ response: The authors thank the reviewer for this observation. We corrected the mentioned mistakes.

(R3.10): Authors have used a well-known dataset; however, they didn’t compare their results with existing studies who have used the same dataset.

Authors’ response: The authors appreciate the reviewer’s comment. We have modified the manuscript based on this comment in Page (lines 273-289) revealing that:

“Moreover, a comparison with previous studies which used the same dataset is conducted, where Giotis et al. [26] proposed a classification system based on color and texture lesion descriptors using the same image database. The automatically extracted features were combined with a set of visual attributes provided by the examining physician. The most valuable features were selected using a majority vote and 81% diagnostic accuracy was reported. The cost of the computation process is high, where the features were generated from images and visual features. In the context of comparison, our proposed method reduces the dimension of feature space, and the feature redundancy, as well as it conduces to better recognition performance with a low computation cost with 100% classification accuracy using 3 features ‘’.

Compared to Hosny et al. [27] who utilized a pre-trained deep convolutional neural network system for automated skin lesion classification of melanoma, nevus, seborrheic keratos with transfer learning and the pre-trained deep neural network with reported 91.18% r average accuracy of the two classes, 91.43% for average accuracy of sensitivity and specificity, and 90.70% for average precision. However, ANN with larger number of hidden layers can affect accuracy and increases the computation cost. On contrary, our proposed method using ANN of 10 hidden layers proved that our approach can adequately select a small number of features, and the performance of the classifiers using this small set of features is improved.”

(R3.11): Figure 3: ROC curve: x-axis title is not correct. It should be 1- Specificity, why x-axis values are from 0-80 why not 0-100?, x-axis and y-axis values should be from 0-1 not from 0-100. In AUC and J values: . should be used instead of ,

Authors’ response: We are appreciating this comment. We have done all the required corrections in Page 7 (figure 3) as follows.

(a)

 (b)

 (c)

(d)

 (e)

(f)

Figure 3. ROC curves and J index corresponding to the selected features for non-melanoma (top line), and melanoma (bottom line) skin lesion, where (a, d) for – AI (asymmetry index) feature, (b, e) for S (skewness) feature, and (c, f) – E (eccentricity) feature.  

Reviewer 4 Report

This work is of research significance, and the proposed method is novel to some extent.  In general, the paper is well written and

can be understood. Some small mistakes are detected as follows:

1) In line 96 of page 3: "C1={T+1,T+2,...,L?1}" may be "C1={T+1,T+2,...,L-1}" 

2) In line 98 of page 3: "the gray level I" should be "the gray level i".

3) Algorithm 1 should be cited by the previous text. The Input and Output of Algorithm 1 should be pointed out. The word "Stop"

should be "End".

4) In line 157 of page 6: "4 Gb" should be "4 GB".

Even though, I suggest that the authors make  a comprehensive grammar check.

Author Response

This work is of research significance, and the proposed method is novel to some extent.  In general, the paper is well written and can be understood.

Authors’ response: We are thankful to the respected reviewer for this encouraging comment and the precious time to provide the feedback. We response to the reviewer's comments as follows

 (R4.1): Some small mistakes are detected as follows:

1) In line 96 of page 3: "C1={T+1,T+2,...,L?1}" may be "C1={T+1,T+2,...,L-1}" 

2) In line 98 of page 3: "the gray level I" should be "the gray level i".

3) Algorithm 1 should be cited by the previous text. The Input and Output of Algorithm 1 should be pointed out. The word "Stop"

should be "End".

4) In line 157 of page 6: "4 Gb" should be "4 GB".

Authors’ response: We are appreciating this comment. We have modified Section (2.1) by deleting the well-established method. In addition, we added some experimental results of the segmentation results on different types of images (benign and malignant) are presented in Figure 1 (below).

 (a1)

(a2)

(b1)

(b2)

(c1)

(c2)

(d1)

(d2)

(e1)

(e2)

(f1)

(f2)

Figure 1. Skin lesions segmentation, index 1 for nevus and 2 for melanoma; (a1), (a2) original RGB image; (b1), (b2) ellipse circumscribed to the lesion; (c1), (c2) elliptical mask; (d1), (d2) elliptical mask enclosing the skin lesion; (e1), (e2) elliptical mask in the gray scale containing the lesion, and (f1), (f2) mask associated with skin lesion (segmented image)

(R4.2): Even though, I suggest that the authors make  a comprehensive grammar check.

Authors’ response: We are thankful to the respected reviewer for giving us the chance to revise our paper and improve its quality. We have revised and corrected our manuscript.

Round 2

Reviewer 1 Report

Unfortunately, the revised manuscript failed to encompass the necessary corrections

There are several significant issues with the followed methodology that were not addressed:

1) Authors address a binary classification problem (nevi versus melanoma) using geometrical features {Assymetry index, Eccentricity, Circularity, Compactness} and

the texture features {Skewness, Kurtosis}.

These features are very simplistic to encode differences among the two classes. Also, geometrical features are highly correlated (they measure the same thing).

e.g. circularity and compactness measure the same; one is the reciprocal of the other, and Assymentry Index and Eccentricity encode the same thing, the deviation from the circular shape again.

Texture features Skewness and Kurtosis are not only elementary and naïve for the classification problem but also very sensitive to the noise. Authors use a data set with many glossy images, and some are highly corrupted from hair. However, highly noisy images should be excluded from the study.

2) The discrimination of melanoma from nevi is a demanding classification task, and proper feature selection is mandatory. However, authors perform feature selection by ranking the “power” of individual features. This option is misleading and does not guarantee the optimum feature vector selection since suffer from the well known inherent problem of nesting.

Roc curves are used in a completely wrong way: Sensitivity is the accuracy of correct classifying Melanoma and Specificity the accuracy of classifying nevus. Figure 3 is unacceptable since authors use different ROC curves for melanoma and nevus.

Finally, in the revised manuscript, authors employ an ANN with ten hidden neurons and claim an accuracy of 100%.  I am afraid that this is the training accuracy of the classifier. Moreover, authors with a rather small data set (170 instances) use a classification model with too many parameters (3 inputs-10 hidden neurons and one output resulting in 40 weights and 11 bias components!!! ). The result is an over-trained model that of course, might give 100% training accuracy but with very low generalization.

Authors have to revise their methodology thoroughly. They should employ more sophisticated features of shape and texture and validate their classification power using robust models such as SVM that can accommodate large feature vectors and are suited for small data sets.

Reviewer 2 Report

Required few Monir correction at line 137 and 138 equation 11